# PRIORITYCUT: OCCLUSION-AWARE REGULARIZATION FOR IMAGE ANIMATION

## ABSTRACT

Image animation generates a video of a source image following the motion of a driving video. Self-supervised image animation approaches do not require explicit pose references as inputs, thus offering large flexibility in learning. State-of-the-art self-supervised image animation approaches mostly warp the source image according to the motion of the driving video, and recover the warping artifacts by inpainting. When the source and the driving images have large pose differences, heavy inpainting is necessary. Without guidance, heavily inpainted regions usually suffer from loss of details. While previous data augmentation techniques such as CutMix are effective in regularizing non-warp-based image generation, directly applying them to image animation ignores the difficulty of inpainting on the warped image. We propose PriorityCut, a novel augmentation approach that uses the top-$k$ percent occluded pixels of the foreground to regularize image animation. By taking into account the difficulty of inpainting, PriorityCut preserves better identity than vanilla CutMix and outperforms state-of-the-art image animation models in terms of the pixel-wise difference, low-level similarity, keypoint distance, and feature embedding distance.

Source Image    Driving Image    Generated Image    Occlusion Mask    PriorityCut Mask    CutMix Image

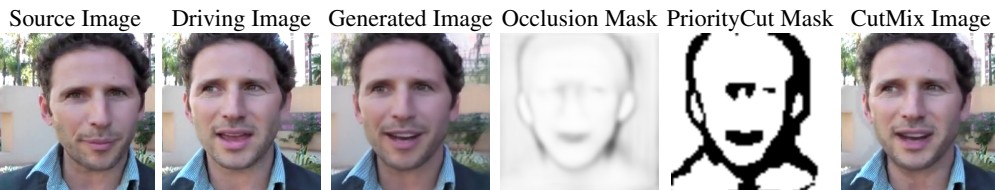

Figure 1: Warp-based image animation warps the source image based on the motion of the driving image and recovers the warping artifacts by inpainting. PriorityCut utilizes the occlusion information in image animation indicating the locations of warping artifacts to regularize discriminator predictions on inpainting. The augmented image has smooth transitions without loss or mixture of context.

## 1 INTRODUCTION

Image animation takes an image and a driving video as inputs and generates a video of the input image that follows the motion of the driving video. Traditional image animation requires a reference pose of the animated object such as facial keypoints or edge maps (Fu et al., 2019; Ha et al., 2019; Qian et al., 2019; Zhang et al., 2019b; Otberdout et al., 2020). Self-supervised image animation does not require explicit keypoint labels on the objects (Wiles et al., 2018; Kim et al., 2019; Siarohin et al., 2019a;b). Without explicit labeling, these approaches often struggle to produce realistic images when the poses between the source and the driving images differ significantly.

To understand this problem, we first look at the typical process of self-supervised image animation approaches. These approaches can be generalized into the following pipeline: (1) keypoint detection, (2) motion prediction, and (3) image generation. Keypoint detection identifies important points in the source image for movement. Motion prediction estimates the motion of the source image based on the driving image. Based on the results of keypoint detection and motion prediction, it warps the source image to obtain an intermediate image that closely resembles the motion of the driving image.

Image generation then recovers the warping artifacts by inpainting. Existing approaches mostly provide limited to no guidance on inpainting. The generator has to rely on the learned statistics to recover the warping artifacts. For instance, First Order Motion Model (Siarohin et al., 2019b) predicts an occlusion mask that indicates where and how much the generator should inpaint. While it has shown significant improvements over previous approaches such as X2Face (Wiles et al., 2018) and Monkey-Net (Siarohin et al., 2019a), it struggles to inpaint realistic details around heavily occluded areas. The occlusion mask does not provide information on how well the generator inpaints.

We propose PriorityCut, a novel augmentation approach that uses the top-$k$ percent occluded pixels of the foreground for consistency regularization. PriorityCut derives a new mask from the occlusion mask and the background mask. Using the PriorityCut mask, we apply CutMix operation (Yun et al., 2019), a data augmentation that cuts and mixes patches of different images, to regularize discriminator predictions. Compared to the vanilla rectangular CutMix mask, PriorityCut mask is flexible in both shape and locations. Also, PriorityCut prevents unrealistic patterns and information loss unlike previous approaches (DeVries & Taylor, 2017; Yun et al., 2019; Zhang et al., 2017). The subtle differences in our CutMix image allow the generator to take small steps in learning, thus refining the details necessary for realistic inpainting. We built PriorityCut on top of First Order Motion Model and experimented on the VoxCeleb (Nagrani et al., 2017), BAIR (Ebert et al., 2017), and Tai-Chi-HD (Siarohin et al., 2019b) datasets. Our experimental results show that PriorityCut outperforms state-of-the-art image animation approaches in *pixel-wise difference*, *low-level similarity*, *keypoint distance*, and *feature embedding distance*.

## 2 RELATED WORK

**Data augmentation** Our work is closely related to patch-based augmentation techniques. Cutout and its variants drop random patches of an image (DeVries & Taylor, 2017; Singh et al., 2018; Chen, 2020). Mixup blends two images to generate a new sample (Zhang et al., 2017). CutMix and its variants cut and mix patches of random regions between images (Takahashi et al., 2019; Yun et al., 2019; Yoo et al., 2020). Yoo et al. (2020) observed that existing patch-based data augmentation techniques either drop the relationship of pixels, induce mixed image contents within an image, or cause a sharp transition in an image. In contrast, we design our augmentation to avoid these issues.

**Image animation** Traditional image animation requires a reference pose of the animated object such as facial keypoints or edge maps (Fu et al., 2019; Ha et al., 2019; Qian et al., 2019; Zhang et al., 2019b; Otberdout et al., 2020). Self-supervised image animation does not require explicit labels on the objects. X2Face (Wiles et al., 2018) uses an embedding network and a driving network to generate images. Kim et al. (2019) used a keypoint detector and a motion generator to predict videos of an action class based on a single image. Monkey-Net (Siarohin et al., 2019a) generates images based on a source image, relative keypoint movements, and dense motion. First Order Motion Model (Siarohin et al., 2019b) extended Monkey-Net by predicting Jacobians in keypoint detection and an occlusion mask. Burkov et al. (2020) achieved pose-identity disentanglement using a big identity encoder and a small pose encoder. Yao et al. (2020) generated images based on optical flow predicted on 3D meshes. These approaches mostly provide limited to no guidance on inpainting. In contrast, our approach utilizes the occlusion information to guide inpainting.

**Advancements in generative adversarial networks** Researchers have proposed different solutions to address the challenges of GANs (Bissoto et al., 2019). Our work is closely related to architectural methods, constraint techniques, and image-to-image translation. Chen et al. (2018) modulated the intermediate layers of a generator by the input noise vector using conditional batch normalization. Kurach et al. (2019) conducted a large-scale study on different regularization and normalization techniques. Some researchers applied consistency regularization on real images (Zhang et al., 2019a), and additionally on generated images and latent variables (Zhao et al., 2020). Researchers also provided local discriminator feedback on patches (Isola et al., 2017) and individual pixels with CutMix regularization (Schonfeld et al., 2020). Our work differs from Schonfeld et al. (2020) in the application domain, mask shape, and mask locations. First, their experiments are on non-warp-based image generation, but we experimented with image animation. Also, their CutMix mask is rectangular and is applied at arbitrary locations. In contrast, our mask shape is irregular and

is applied to heavily occluded areas. In Section 3.1, we discuss the implications of directly applying the vanilla CutMix to image animation.

# 3 METHODOLOGY

The core of our methodology is to guide the model to gradually learn the crucial parts in inpainting. We first summarize the base architecture we built upon. Then, we introduce per-pixel discriminator feedback and its importance in image animation. After that, we discuss the limitations of directly applying existing patch-based data augmentation on image animation. Lastly, we illustrate how the limitations of existing data augmentation techniques inspired the design of our approach.

## 3.1 BACKGROUND

**First Order Motion Model** We built our architecture on top of First Order Motion Model (Siarohin et al., 2019b), a state-of-the-art model on image animation. First Order Motion Model consists of a motion estimation module and an image generation module. The motion estimation module takes as inputs a source image $\mathbf{S}$ and a driving image $\mathbf{D}$, and predicts a dense motion field $\hat{\mathcal{T}}_{\mathbf{S}\leftarrow\mathbf{D}}$ and an occlusion mask $\hat{\mathcal{O}}_{\mathbf{S}\leftarrow\mathbf{D}}$. The image generation module warps the source image based on the dense motion field $\hat{\mathcal{T}}_{\mathbf{S}\leftarrow\mathbf{D}}$ and recovers warping artifacts by inpainting the occluded parts of the source image. For details of individual modules, see Section A of the appendix.

**Per-pixel discriminator feedback** In image recovery, the generator needs to maintain both the global and local realism. Existing image animation techniques either provide no clues (Wiles et al., 2018; Siarohin et al., 2019a) or limited clues like occlusion map (Siarohin et al., 2019b; Yao et al., 2020) to guide inpainting. A reenacted image can share a similar pose as the driving image (*global realism*), but the subtle texture or geometry differences can affect the perspective of identity (*local realism*). To address this issue, we adapted the U-Net discriminator architecture (Schonfeld et al., 2020) to provide both global and per-pixel discriminator feedback. A U-Net discriminator $D^U$ consists of an encoder $D^U_{enc}$ and a decoder $D^U_{dec}$. The encoder serves the same purpose as an ordinary discriminator, which predicts if an image is real or fake (*global realism*). The decoder predicts if individual pixels are real or fake (*local realism*). The U-Net discriminator uses skip connections to feed information between matching resolutions of the encoder and the decoder. Per-pixel discriminator feedback is especially important in warping-based image animation techniques. After warping, some regions of the warped image resemble the source image less than the other regions, and recovering the artifacts around those regions requires relatively more inpainting effort. Per-pixel feedback helps the generator learn precisely *where* and *how much* to improve during inpainting. For details of our architectures, see Section B of the appendix.

**Consistency regularization** While per-pixel discriminator feedback provides fine-grained feedback, there is no guarantee on consistent predictions. Researchers have demonstrated the effectiveness of CutMix (Yun et al., 2019) in regularizing the U-Net decoder (Schonfeld et al., 2020). However, directly applying vanilla CutMix to image animation has a few limitations. First, unlike Schonfeld et al. (2020) that experimented on non-warp-based image generation, image warping creates a gradient of inpainting difficulty on a single image. Applying CutMix at arbitrary locations makes it difficult for the generator to focus on improving the heavily occluded areas. Also, there is a mismatch between the mask shape and the task nature. In image animation, there can be multiple occluded regions of irregular shapes. Simply using a single rectangular mask for CutMix like Schonfeld et al. (2020) does not reflect the reality of the task. Finally, regularizing the discriminator with vanilla CutMix can provide only partial per-pixel feedback on image restoration. Yoo et al. (2020) suggested that good augmentation techniques should not have sharp transitions like CutMix (Yun et al., 2019), mixed image contents within an image patch like Mixup (Zhang et al., 2017), or losing the relationships of pixels like Cutout (DeVries & Taylor, 2017). In the vanilla CutMix image, part of the image context is replaced by that of another image. Mixing per-pixel feedback may confuse the generator on restoring the artifacts. To fully utilize per-pixel discriminator feedback, an augmentation mask should closely reflect the tasks a model is trying to learn.

| Occluded Pixels | Source Image | Driving Image | Generated Image | Occlusion Mask | Background Mask | PriorityCut Mask | CutMix Image |
|---|---|---|---|---|---|---|---|

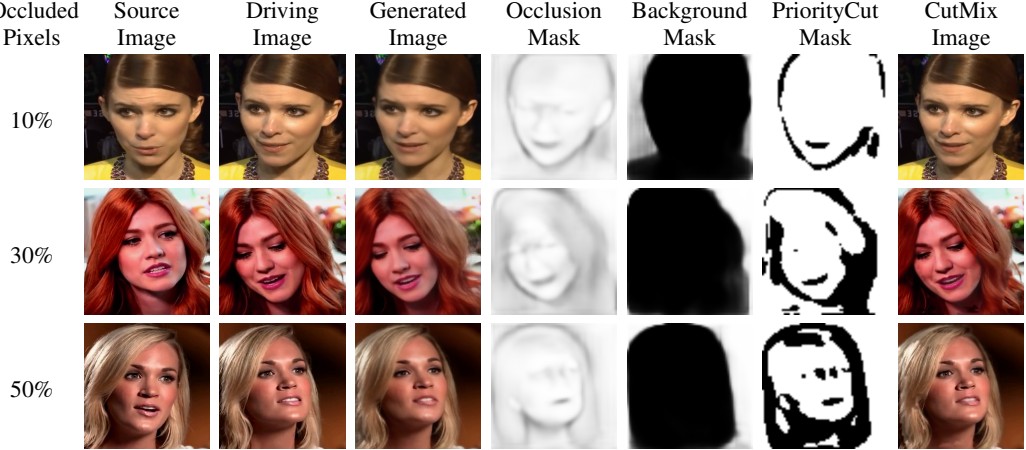

Figure 2: Illustration of deriving PriorityCut masks from occlusion and background masks.

| Source | Driving | Vanilla CutMix Reconstruction | Vanilla CutMix Feedback | PriorityCut Reconstruction | PriorityCut Feedback |
|---|---|---|---|---|---|

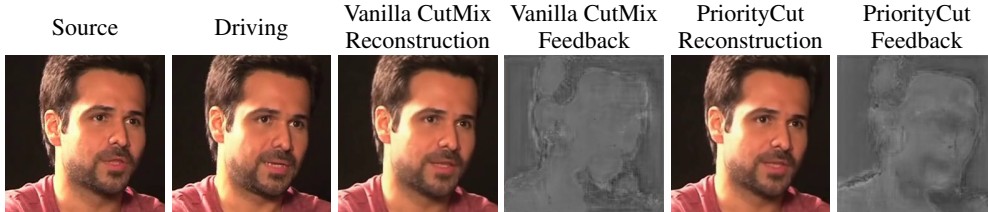

Figure 3: Comparison of per-pixel discriminator feedback between vanilla CutMix and PriorityCut.

## 3.2 PRIORITYCUT

Our approach is based on two key observations. One observation is that occlusion in warping-based image animation reflects the intensity of artifacts that need to be recovered. Another observation is that heavy occlusion can happen on both the foreground and the background. To recover the artifacts effectively, the generator should focus its learning on *heavily occluded areas* and the *main object*.

Based on the above observations, we propose PriorityCut, a novel augmentation that uses the top-$k$ percent occluded pixels of the foreground as the CutMix mask. Figure 2 illustrates the derivation of PriorityCut masks from occlusion and background masks. Suppose $\mathcal{M}_{bg}$ is an alpha background mask predicted by the dense motion network, ranging between 0 and 1. We first suppress the uncertain pixels of the alpha background mask $\mathcal{M}_{bg}$ to obtain a binary background mask $\hat{\mathcal{M}}_{bg}$. $\hat{\mathcal{M}}_{bg}$ corresponds to the background mask predicted by the dense motion network with high confidence. The occlusion map $\hat{\mathcal{O}}_{\mathbf{S}\leftarrow\mathbf{D}} \in [0,1]^{H\times W}$ is an alpha mask, with 0 being fully occluded and 1 being not occluded. Equation 1 utilizes $\hat{\mathcal{M}}_{bg}$ to compute the occlusion map of the foreground $\hat{\mathcal{O}}_{fg}$:

$$\hat{\mathcal{O}}_{fg} = \hat{\mathcal{M}}_{bg} + (1 - \hat{\mathcal{M}}_{bg}) \odot \hat{\mathcal{O}}_{\mathbf{S}\leftarrow\mathbf{D}} \tag{1}$$

where $\odot$ denotes the Hadamard product. It retains only the foreground portions of the occlusion masks shown in Figure 2, which are also alpha masks. Given a percentile $k$, we denote the PriorityCut mask $\mathcal{M}_{pc}$ as the top-$k$ percent occluded pixels of the foreground $\hat{\mathcal{O}}_{fg}$. Following Yun et al. (2019), we randomize the values of $k$ in our experiments. Equation 2 utilizes the PriorityCut mask to perform CutMix between the real images $x$ and the generated images $x'$. To avoid sharp transitions, PriorityCut performs CutMix on the driving image $\mathbf{D}$ and its reconstruction $\hat{\mathbf{D}}$.

$$\text{mix}(x, x', \mathcal{M}_{pc}) = \mathcal{M}_{pc} \odot x + (1 - \mathcal{M}_{pc}) \odot x' \tag{2}$$

In Figure 2, the CutMix images look almost identical to the driving or the generated images with only subtle differences in fine details. PriorityCut always assigns the fake pixels to locations where there are large changes in motion, creating incentives for the generator to improve. For example, borders,

edges, in-between regions of distinct objects (e.g. face, mic, wall), or parts of objects (e.g. hair, eyes, nose, mouth). The design philosophy of PriorityCut follows that of CutBlur (Yoo et al., 2020). The augmented images have no sharp transitions, mixed image contents, or loss of the relationships of pixels. PriorityCut also adds another degree of flexibility to the mask shapes. The discriminator can no longer rely on a rectangular area like the vanilla CutMix to predict where the real and fake pixels concentrate at. This encourages the discriminator to learn properly the locations of the real and fake pixels. Figure 3 compares the per-pixel discriminator feedback between PriorityCut and vanilla CutMix. PriorityCut helps the discriminator learn clear distinctions between real and fake pixels around locations with large changes in motion. In contrast, vanilla CutMix helps the discriminator learn only vague estimations. In Section 4.3, we compare PriorityCut with applying vanilla CutMix at arbitrary locations.

### 3.3 Training losses

We followed previous works (Siarohin et al., 2019b; Schonfeld et al., 2020) to use a combination of losses. The U-Net discriminator loss $\mathcal{L}_{D^U}$ consists of the adversarial losses of the U-Net encoder $\mathcal{L}_{D^U_{enc}}$ and the U-Net decoder $\mathcal{L}_{D^U_{dec}}$, and the consistency regularization loss $\mathcal{L}^{cons}_{D^U_{dec}}$. The consistency regularization loss regularizes the U-Net decoder output on the CutMix image and the CutMix between the U-Net decoder outputs on real and fake images. The generator loss $\mathcal{L}_G$ consists of the reconstruction loss $\mathcal{L}_{rec}$ based on activations of the pre-trained VGG-19 network (Simonyan & Zisserman, 2014), the equivariance loss $\mathcal{L}_{equiv}$ on local motion approximation to encourage consistent keypoint predictions, the adversarial loss $\mathcal{L}_{adv}$, and the feature matching loss $\mathcal{L}_{feat}$ similar to that of pix2pixHD (Wang et al., 2018). For more details, refer to Section B in the appendix.

## 4 Experiments

### 4.1 Experimental setup

**Datasets**    We followed Siarohin et al. (2019b) to preprocess high-quality videos on the following datasets and resized them to $256 \times 256$ resolution: the VoxCeleb dataset (Nagrani et al., 2017) (18,398 training and 512 testing videos after preprocessing); the Tai-Chi-HD dataset (Siarohin et al., 2019b) (2,994 training and 285 testing video chunks after preprocessing); the BAIR robot pushing dataset (Ebert et al., 2017) (42,880 training and 128 testing videos).

**Evaluation protocol**    We followed Siarohin et al. (2019b) to quantitatively and qualitatively evaluate video reconstruction. For video reconstruction, we used the first frame of the input video as the source image and each frame as the driving image. We evaluated the reconstructed videos against the ground truth videos on the following metrics: *pixel-wise differences* ($\mathcal{L}_1$); *PSNR*, *SSIM*, and their masked versions (*M-PSNR*, *M-SSIM*); *average keypoint distance (AKD)*, *missing keypoint rate (MKR)*, and *average Euclidean distance (AED)* of feature embeddings detected by third-party tools.

For details on dataset preprocessing and metric computation, refer to Section C in the appendix.

### 4.2 Comparison with state-of-the-art

We quantitatively and qualitatively compared PriorityCut with state-of-the-art self-supervised image animation methods with publicly available implementations.

- **X2Face**. The reenactment system with an embedding and a driving network (Wiles et al., 2018).

- **Monkey-Net**. The motion transfer framework based on a keypoint detector, a dense motion network, and a motion transfer generator (Siarohin et al., 2019a).

- **First Order Motion Model**. The motion transfer network that extends Monkey-Net by estimating affine transformations for the keypoints and predicting occlusion for inpainting (Siarohin et al., 2019b). We compared two versions of First Order Motion Model. The baseline model (FOMM) corresponds to the one in their published paper. The adversarial model (FOMM+) is a concurrent

| Model | $\mathcal{L}_1 \downarrow$ | PSNR ↑ | | | SSIM ↑ | | | AKD ↓ | AED ↓ |
|---|---|---|---|---|---|---|---|---|---|
| | | All | Salient | ¬ Salient | All | Salient | ¬ Salient | | |
| X2Face | 0.0739±2e-4 | 19.13±0.02 | 20.04±0.02 | 30.65±0.04 | 0.625±6e-4 | 0.681±5e-4 | 0.944±2e-4 | 6.847±4e-3 | 0.3664±2e-3 |
| Monkey-Net | 0.0477±1e-4 | 22.47±0.02 | 23.29±0.02 | 34.43±0.04 | 0.730±5e-4 | 0.769±4e-4 | 0.962±2e-4 | 1.892±4e-3 | 0.1967±8e-4 |
| FOMM | 0.0413±9e-5 | 24.28±0.02 | 25.19±0.02 | 36.19±0.04 | 0.791±4e-4 | 0.825±4e-4 | 0.969±2e-4 | 1.290±2e-3 | 0.1324±6e-4 |
| FOMM+ | 0.0409±9e-5 | 24.26±0.02 | 25.17±0.02 | 36.26±0.04 | 0.790±4e-4 | 0.822±4e-4 | 0.970±1e-4 | 1.305±2e-3 | 0.1339±6e-4 |
| Ours | 0.0401±9e-5 | 24.45±0.02 | 25.35±0.02 | 36.45±0.04 | 0.793±4e-4 | 0.826±2e-4 | 0.970±1e-4 | 1.286±2e-3 | 0.1303±6e-4 |

Table 1: Comparison with state-of-the-art for approaches for video reconstruction on VoxCeleb.

| Model | $\mathcal{L}_1 \downarrow$ | PSNR ↑ | | | SSIM ↑ | | | AKD ↓ | MKR ↓ | AED ↓ |
|---|---|---|---|---|---|---|---|---|---|---|
| | | All | Salient | ¬ Salient | All | Salient | ¬ Salient | | | |
| X2Face | 0.0729±3e-4 | 18.16±0.02 | 21.08±0.02 | 22.24±0.02 | 0.580±1e-3 | 0.858±3e-4 | 0.734±1e-3 | 14.89±8e-2 | 0.175±1e-3 | 0.2441±6e-4 |
| Monkey-Net | 0.0691±3e-4 | 18.89±0.03 | 22.02±0.03 | 22.70±0.04 | 0.599±2e-3 | 0.867±3e-4 | 0.742±1e-3 | 11.40±7e-2 | 0.060±7e-4 | 0.2319±7e-4 |
| FOMM | 0.0569±2e-4 | 21.29±0.03 | 24.65±0.03 | 25.18±0.04 | 0.651±2e-3 | 0.891±3e-4 | 0.771±1e-3 | 6.87±6e-2 | 0.038±5e-4 | 0.1657±6e-4 |
| FOMM+ | 0.0555±2e-4 | 21.35±0.03 | 24.74±0.03 | 25.21±0.04 | 0.654±2e-3 | 0.893±3e-4 | 0.772±1e-3 | 6.73±6e-2 | 0.032±4e-4 | 0.1647±6e-4 |
| Ours | 0.0549±2e-4 | 21.54±0.03 | 24.98±0.03 | 25.33±0.04 | 0.653±2e-3 | 0.896±3e-4 | 0.768±1e-3 | 6.78±6e-2 | 0.030±4e-4 | 0.1629±6e-4 |

Table 2: Comparison with state-of-the-art for approaches for video reconstruction on Tai-Chi-HD.

work with an adversarial discriminator. Since its authors have released[1] both models, we evaluated the baseline model and additionally the adversarial model.

- **Ours**. Our extension of First Order Motion Model with U-Net discriminator to provide per-pixel discriminator feedback and PriorityCut to regularize inpainting.

**Quantitative comparison** Table 1, 2, and 3 show the quantitative comparison results of video reconstruction on the VoxCeleb, BAIR, and Tai-Chi-HD datasets, respectively. For all tables, the down arrows indicate that lower values mean better results, and vice versa. We show the 95% confidence intervals, highlight the best results in bold and underline the second-best. For variants of the baseline model that do not produce the best or the second best results, the red and green texts indicate worse and better results than the baseline, respectively. This serves a similar purpose as the ablation study, indicating the effectiveness of certain components in improving the baseline. PriorityCut outperforms state-of-the-art models in every single metric for VoxCeleb and BAIR, and in most of the metrics for Tai-Chi-HD. Note that adversarial training alone (FOMM+) does not always guarantee improvements, as highlighted in red for VoxCeleb.

**Qualitative comparison** Figure 4 shows the qualitative comparison for the VoxCeleb and BAIR datasets. The color boxes highlight the noticeable differences between the results of different models.

For the VoxCeleb dataset, X2Face produces slight to heavy distortions on the face, depending on the pose angles. Monkey-Net either fails to follow the pose angles or struggles to preserve the identity of the source image. FOMM follows closely the pose angles, but it struggles to inpaint the subtle details. For instance, the corner of the right eye extends all the way to the hair (frame 1; frame 3 below), the

| Model | $\mathcal{L}_1 \downarrow$ | PSNR ↑ | SSIM ↑ |
|---|---|---|---|
| X2Face | 0.0419±5e-4 | 21.3±0.1 | 0.831±2e-3 |
| Monkey-Net | 0.0340±4e-4 | 23.1±0.1 | 0.867±2e-3 |
| FOMM | 0.0292±4e-4 | 24.8±0.1 | 0.889±1e-3 |
| Ours | 0.0276±3e-4 | 25.3±0.1 | 0.894±1e-3 |

Table 3: Comparison with state-of-the-art for approaches for video reconstruction on BAIR.

hair on the background (frame 2), and the left eye in a polygon shape (frame 3 top). FOMM+ either amplifies the artifacts (frames 1 and 3) or is uncertain about the texture (frame 2). In contrast, PriorityCut maintains a clear distinction between the right eye and the hair (frame 1; frame 3 below), inpaints the left eye in the ellipse shape (frame 3 top), and has high confidence in the texture (frame 2).

---

[1]https://github.com/AliaksandrSiarohin/first-order-model

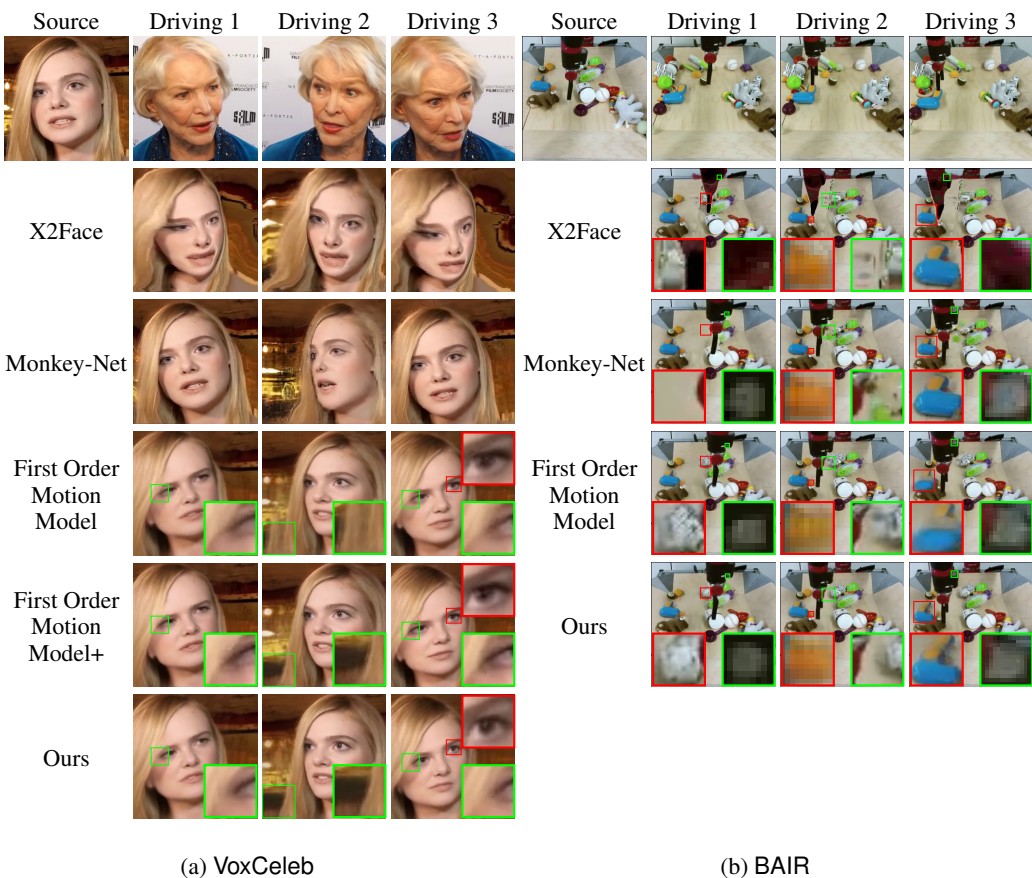

(a) VoxCeleb

(b) BAIR

Figure 4: Qualitative comparison of state-of-the-art approaches for image animation.

For the BAIR dataset, X2Face produces trivial warping artifacts. Monkey-Net either erases the object (frame 1 left) or introduces extra artifacts (frame 2 right). FOMM is uncertain about the texture (frame 1 left; frames 2 and 3) or the geometry (frame 1 right). In contrast, PriorityCut inpaints both realistic texture and geometry. Note that the blue and yellow object in the third frame is stretched due to image warping. PriorityCut recovers the texture while FOMM splits the object into two parts.

For additional qualitative comparison, refer to Section D of the appendix.

### 4.3 ABLATION STUDY

To validate the effects of each proposed component, we evaluated the following variants of our model on video reconstruction. *Baseline*: the published First Order Motion Model used in their paper; *Adv*: the concurrent work of First Order Motion Model with a global discriminator; *U-Net*: the architecture of the global discriminator extended to the U-Net architecture; *PriorityCut*: our proposed approach that uses the top-$k$ percent occluded pixels of the foreground as the CutMix mask.

**Quantitative ablation study**  Table 4 quantitatively compares the results of video reconstruction on the VoxCeleb dataset (Nagrani et al., 2017). First, adversarial training improves only the $\mathcal{L}_1$ distance and the non-salient parts, but worsens other metrics. U-Net discriminator improves $\mathcal{L}_1$ by a margin with better *AKD* as a positive side bonus, at the cost of further degraded *AED*. We experimented with adding either PriorityCut or vanilla CutMix on top of the U-Net architecture. After adding PriorityCut, the full model outperforms the baseline model in every single metric. In particular, the improvement of *AED* shows the effectiveness of PriorityCut in guiding the model to inpaint realistic facial features. However, vanilla CutMix pushes the generator to optimize only the pixel values, at the cost of significant degradation in keypoint distance (*AKD*) and identity preservation (*AED*).

| Architecture | $\mathcal{L}_1 \downarrow$ | PSNR ↑ | | | SSIM ↑ | | | AKD ↓ | AED ↓ |
|---|---|---|---|---|---|---|---|---|---|
| | | All | Salient | ¬ Salient | All | Salient | ¬ Salient | | |
| Baseline | 0.0413±9e-5 | 24.28±0.02 | 25.19±0.02 | 36.19±0.04 | 0.791±4e-4 | 0.825±4e-4 | 0.969±2e-4 | 1.290±2e-3 | 0.1324±6e-4 |
| + Adv | 0.0409±9e-5 | 24.26±0.02 | 25.17±0.02 | 36.26±0.04 | 0.790±4e-4 | 0.822±2e-4 | **0.970**±1e-4 | 1.305±2e-3 | 0.1339±6e-4 |
| + U-Net | 0.0401±9e-5 | 24.34±0.02 | 25.29±0.02 | 36.31±0.04 | 0.791±4e-4 | 0.824±4e-4 | 0.969±2e-4 | **1.278**±2e-3 | 0.1347±6e-4 |
| + PriorityCut | 0.0401±9e-5 | 24.45±0.02 | 25.35±0.02 | **36.45**±0.04 | **0.793**±4e-4 | **0.826**±4e-4 | **0.970**±1e-4 | 1.286±2e-3 | **0.1303**±6e-4 |
| + CutMix | **0.0394**±9e-5 | **24.51**±0.02 | **25.45**±0.02 | 36.42±0.02 | 0.792±4e-4 | **0.826**±4e-4 | 0.969±2e-4 | 1.295±2e-3 | 0.1365±6e-4 |

Table 4: Quantitative ablation study for video reconstruction on VoxCeleb.

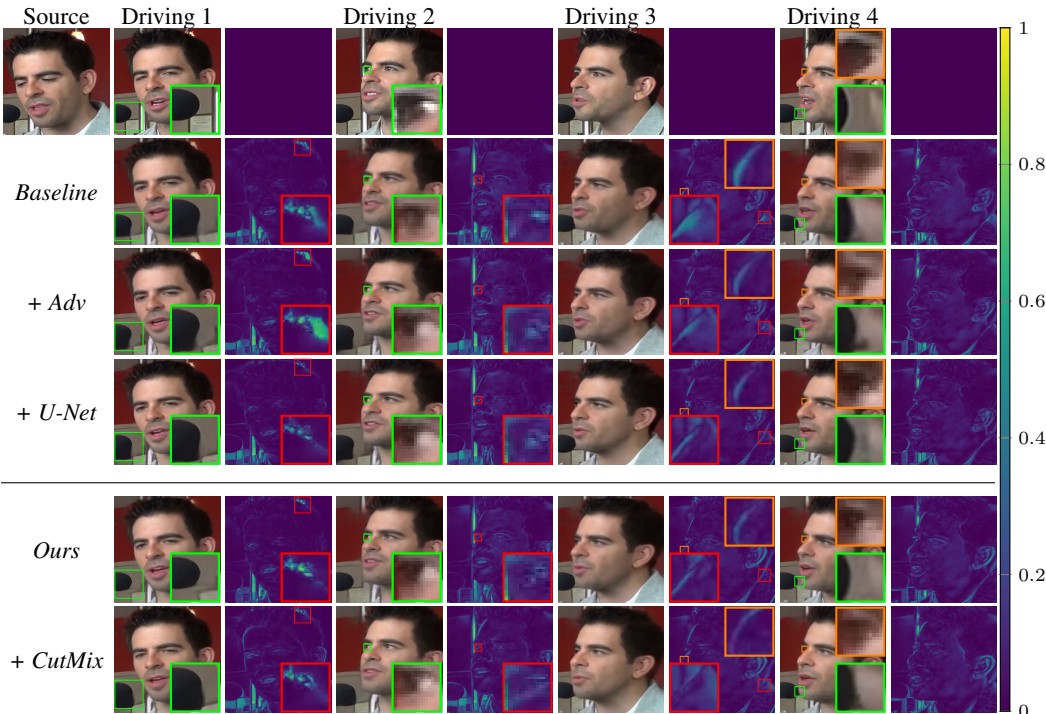

Figure 5: Qualitative ablation study for video reconstruction on VoxCeleb.

**Qualitative ablation study** Figure 5 qualitatively compares video reconstruction on the VoxCeleb dataset (Nagrani et al., 2017). The first row is the ground truth. The heatmaps illustrate the differences between the ground truth and the reconstructed frames. The color boxes highlight the noticeable differences between architectures. First, adversarial training *amplifies the texture* (right eyes of frames 2 and 4; artifacts in the heatmaps of frames 1 and 2). Per-pixel discriminator feedback *produces more precise texture* than the adversarial model (heatmaps in frames 1 and 3; right eye of frame 4). Similar to quantitative ablation study, we qualitatively compared the effects of adding either PriorityCut or vanilla CutMix on top of the U-Net architecture. PriorityCut is *sensitive to areas with large changes in motion*. Among different architectures, PriorityCut is the only one that maintains the mic shape (frame 1) and the distance between the mic and the mouth (frame 4). Also, the heatmaps of frames 2 and 3 for PriorityCut resemble the ground truth the most. For vanilla CutMix, the mic shape problem persists (frame 1), the right eye shows ambiguous texture (frame 4 above) and the gap between the mic and the mouth is closed (frame 4 below). Most importantly, all heatmaps of vanilla CutMix show trivial differences in texture around the edge of the right face. These suggest that vanilla CutMix *struggles in distinguishing between foreground and background around locations with large motions*, since its augmentation mask was not designed to handle motions. Overall, qualitative ablation study results show the effectiveness of PriorityCut in capturing the subtle shape and texture.

## 5 DISCUSSION

This section summarizes the key observations and findings of our work.

**Limitations of warp-based image animation**   Existing warp-based image animation techniques recover the warping artifacts by inpainting. We observed that large pose differences in image animation often critically influence identity preservation. Without proper guidance, the generator usually struggles at recovering the warping artifacts at locations with large motions. To address this challenge, we proposed PriorityCut to regularize image animation based on inpainting difficulty, capturing the aspects related to large changes in motion. Our experimental results with solid baselines and diverse datasets show that PriorityCut outperforms state-of-the-art models in identity preservation.

**Limitations of regularizing image animation**   While Schonfeld et al. (2020) demonstrated the effectiveness of vanilla CutMix on non-warp-based image generation, we observed that directly applying vanilla CutMix to image animation ignores the inpainting difficultly with augmentation masks irrelevant to motions, and provides only partial per-pixel feedback on image restoration (Section 3.1). Our comparisons with vanilla CutMix in image animation (Section 4.3) support our observations and reveal contradictory findings to that of Schonfeld et al. (2020): directly applying vanilla CutMix to image animation compromises crucial image animation properties such as pose and identity to pursue pixel realism. Adversarial training, being an unsupervised approach as vanilla CutMix, faces similar trade-offs when applied to image animation. To address this challenge, PriorityCut takes motion into account and redefines the CutMix augmentation mask to supervise training on the difficult parts of inpainting. The all-round realism of PriorityCut is attributable to the tight coupling between its novel design and the nature of image animation. Our findings substantiate those of Yoo et al. (2020): an augmentation mask closely related to the task nature plays a significant role in effective learning.

**Potential applications of PriorityCut**   One limitation of PriorityCut is the dependency on an occlusion mask and a background mask. Only state-of-the-art image animation approaches use these masks (Kim et al., 2019; Siarohin et al., 2019b; Burkov et al., 2020; Yao et al., 2020). However, we anticipate any warp-based image animation approaches can adopt PriorityCut with proper modifications. In addition to image animation, we expect PriorityCut to be widely applicable to any research areas involve image warping, occlusion, motion or optical flow estimation such as facial expression and body pose manipulation, image inpainting, and video frame interpolation.

## 6 CONCLUSION

We proposed PriorityCut, a novel augmentation approach that captures the crucial aspects related to large changes in motion to address the identify preservation problem in image animation. PriorityCut outperforms state-of-the-art image animation models in terms of the pixel-wise difference, low-level similarity, keypoint distance, and feature embedding distance. Our experimental results demonstrated the effectiveness of PriorityCut in achieving all-round realism and confirmed the significance of augmentation mask in balanced learning.

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
