# OpenReview forum: "PriorityCut: Occlusion-aware Regularization for Image Animation"
_ICLR.cc/2021/Conference — Reject_

### Official Review · AnonReviewer1 · 2020-10-26
**The paper discusses a method to slightly improve existing animation techniques using a U-net discriminator and a mask for inpainting regularization**

**Rating:** 2
**Confidence:** 5

**Review:**

**1. Presentation and clarity**

I believe the paper is very poorly structured, does not introduce related work properly, contains many unclear points in the presentation, making it almost impossible for the reader to grasp key ideas without reading at least three related works on which this paper is heavily based. Until one consults (Schonfeld et al., 2020) it is not clear what the per-pixel feedback means, it's also not clear what is inpainting regularizations. If we look at section 3.3 we might get confused what is the disciminator encoder and decoder, and how the losses are used to train them. These details are fleshed out in the appendix and in the prior work.

I believe the flaw in presentation is due to weak originality of the paper. It borrows heavily from (Schonfeld et al., 2020) and (Siarohin et al., 2019b) and hence does not have much of original content.

**2. Originality**

The paper replaces the discriminator of (Siarohin et al., 2019b) with the one presented in (Schonfeld et al., 2020)  supervising it with the images generated with the priority-cut scheme. The latter uses the occlusion mask of (Siarohin et al., 2019b) to generate a new image by combining the generated image with the driving image, attempting to show the discriminator which pixels require further attention.

In my opinion, these ideas are quite marginal and do not contain any interest for community. The paper re-uses already existing and well working techniques, such as the whole framework of (Siarohin et al., 2019b).

**3. Results**

By looking at the results I cannot see significant differences compared with (Siarohin et al., 2019b). The artifacts present in the first order motion model are present here, so no noticeable improvement overall. The reader will be able to notice something only if they zoom into the figures and compare very small details. To facilitate such behavior, the authors increased the regions in which such details might be noticeable. Even with such visualization, I have to admit, the differences are insignificant. This observation is supported by the numerical results as well. The improvement over FOMM is 0.0012 in terms of L1 on VoxCeleb and 0.002 on Tai-Chi-HD and 0.0016 on BAIR.

**4. Rating**

I believe, poor presentation alone is a sufficient reason to recommend rejection. If we set aside it for a moment, we notice that the presented ideas are simple adaptations from previously published papers and results do not show necessary improvement.

---

> ### Author Response · Authors · 2020-11-19
> **Comments for Review 1 (parts 1/2)**
>
> **Q1. Presentation and clarity**
>
> > I believe the paper is very poorly structured, does not introduce related work properly, contains many unclear points in the presentation, making it almost impossible for the reader to grasp key ideas without reading at least three related works on which this paper is heavily based. Until one consults (Schonfeld et al., 2020) it is not clear what the per-pixel feedback means, it's also not clear what is inpainting regularizations. If we look at section 3.3 we might get confused what is the disciminator encoder and decoder, and how the losses are used to train them. These details are fleshed out in the appendix and in the prior work.
>
> **A1.** We understand that a proper introduction of related work is important to grasp the key ideas. Due to space constraints, we provided the necessary details in the appendix. To improve the readability, we have clarified per-pixel feedback in Section 3.1, inpainting regularization in Figure 1, and the discriminator encoder and decoder in both Sections 3.1 and 3.3.
>
> **Q2. Originality**
> > In my opinion, these ideas are quite marginal and do not contain any interest for community. The paper re-uses already existing and well working techniques, such as the whole framework of (Siarohin et al., 2019b).
>
> **A2.** It is well-known that warping-based image animation techniques struggle to perform when there is large motion, and well-working techniques such as First Order Motion Model (Siarohin et al., 2019b) are not exceptions. We carefully selected source and driving images of large pose differences to demonstrate in our qualitative evaluations that they struggle to handle subtle shapes and textures. While another technique, U-Net GAN with CutMix regularization (Schonfeld et al., 2020) was also proven to be well working in their paper, Section 3.1 discusses exactly why it is not directly applicable to image animation and we have demonstrated in our quantitative ablation study (Table 4) that it optimizes only the pixel values, at the cost of significant degrade in keypoint distance (AKD) and identity preservation (AED). The takeaway in our experiments is that well working techniques are not necessarily one solution for all problems and may not be directly applicable to other applications. We have revised Section 5 Discussion to reflect the key messages accordingly.

---

> ### Author Response · Authors · 2020-11-19
> **Comments for Review 1 (parts 2/2)**
>
> **Q3. Results**
>
> > By looking at the results I cannot see significant differences compared with (Siarohin et al., 2019b). The artifacts present in the first order motion model are present here, so no noticeable improvement overall. The reader will be able to notice something only if they zoom into the figures and compare very small details. To facilitate such behavior, the authors increased the regions in which such details might be noticeable. Even with such visualization, I have to admit, the differences are insignificant. This observation is supported by the numerical results as well. The improvement over FOMM is 0.0012 in terms of L1 on VoxCeleb and 0.002 on Tai-Chi-HD and 0.0016 on BAIR.
>
> **A3.** The degree of improvement is consistent with state-of-the-art techniques in both image animation and image inpainting literature published in top machine learning conferences.
>
> **Image Animation**
>
> For image animation on VoxCeleb, PriorityCut improved the L1 distance by 0.0012 compared to FOMM while Tripathy et al., 2021 [1] worsened the MSE (which also measures pixel distances) by 0.006 compared to FOMM and is the same as X2Face. PriorityCut improved the PSNR by 0.17 while Yao et al., 2020 [2] improved the PSNR by 0.212. PriorityCut improved both identity and pose while Burkov, Egor, et al., 2020 [3] showed a trade-off between identity and pose. For image animation on Tai-Chi-HD, PriorityCut improved the AKD by 0.09 compared to FOMM while Jeon, Subin, et al., 2020 [4] worsened the AKD by 1.27 compared to FOMM and is only on par with MonkeyNet.
>
> **Image Inpainting**
>
> For image inpainting, PriorityCut improved the PSNR and the SSIM in the ranges of 0.15 - 0.5 and 0.001 - 0.005, respectively. Liu, Guilin, et al., 2018 [5] improved the PSNR and the SSIM in the ranges of 0.1 - 0.55 and 0 - 0.008, respectively. Zheng, Chuanxia, et al., 2019 [6] improved the PSNR by 0.74. Xiong, Wei, et al., 2019 [7] improved the PSNR by 0.6 and the SSIM by 0.003. Xie, Chaohao, et al., 2019 [8] improved the PSNR and the SSIM in the ranges of 0.19 - 0.42 and 0.002 - 0.007, respectively.
>
> For visualization, we have included videos corresponding to the figures in the paper with noticeable improvements in the supplementary materials.
>
> **References**
>
> [1] Tripathy, Soumya, Juho Kannala, and Esa Rahtu. "FACEGAN: Facial Attribute Controllable rEenactment GAN." Proceedings of the Workshop on Applications of Computer Vision. 2021.
>
> [2] Yao, Guangming, et al. "Mesh Guided One-shot Face Reenactment Using Graph Convolutional Networks." Proceedings of the 28th ACM International Conference on Multimedia. 2020.
>
> [3] Burkov, Egor, et al. "Neural Head Reenactment with Latent Pose Descriptors." Proceedings of the IEEE/CVF Conference on Computer Vision and Pattern Recognition. 2020.
>
> [4] Jeon, Subin, et al. "Cross-Identity Motion Transfer for Arbitrary Objects through Pose-Attentive Video Reassembling." Proceedings of the 16th European Conference on Computer Vision. 2020.
>
> [5] Liu, Guilin, et al. "Image inpainting for irregular holes using partial convolutions." Proceedings of the European Conference on Computer Vision (ECCV). 2018.
>
> [6] Zheng, Chuanxia, Tat-Jen Cham, and Jianfei Cai. "Pluralistic image completion." Proceedings of the IEEE Conference on Computer Vision and Pattern Recognition. 2019.
>
> [7] Xiong, Wei, et al. "Foreground-aware image inpainting." Proceedings of the IEEE conference on computer vision and pattern recognition. 2019.
>
> [8] Xie, Chaohao, et al. "Image inpainting with learnable bidirectional attention maps." Proceedings of the IEEE International Conference on Computer Vision. 2019.

---

### Official Review · AnonReviewer2 · 2020-10-28

**Rating:** 5
**Confidence:** 2

**Review:**

This paper proposes PRIORITYCUT, where the authors create an occlusion mask based on predicted background map and occlusion map. The occlusion mask indicates which pixels have heavy motions and most of these pixels are on the edges.  Next, the authors utilize this map to create the reconstructed driving image. The benefits are that the discriminator could totally focus on the occluded regions and improve the generating effects.

Comments:
The paper mainly combines the CutMix and First Order Motion Model, which is pretty interesting. However, I have several concerns :

Novelty:  The paper seems to combine two methods together. The network comes from First Order Motion Model while the PRIORITYCUT is built on top of CutMix. After reading the paper, my feeling is that the author slightly changes the formula in CutMix and use this trick to get better reconstruction results. However, I don’t think this paper brings too much new knowledge.

Experiments: in equation 2, there is the top-k number. Which k do you choose in your experiments?  I do see an ablation study in supp but it seems to be an evaluation of masked PSNR and SSIM, not the choice of k. I think it would be helpful to add this. To be specific, does top-k generalize CutMix? For example, if K = 0, it is CutMix?

The good point is that the authors outperform the state of the art. This is a very good point since while the proposed priority cut is simple, it works.


Conclusion: Overall, I think this paper proposes an interesting idea and shows good results. However, due to lack of creativity and inadequate ablation study, I rate it below the acceptance bar.

---

> ### Author Response · Authors · 2020-11-19
> **Comments for Review 2**
>
> **Q1. New knowledge**
>
> > The paper seems to combine two methods together. The network comes from First Order Motion Model while the PRIORITYCUT is built on top of CutMix. After reading the paper, my feeling is that the author slightly changes the formula in CutMix and use this trick to get better reconstruction results. However, I don’t think this paper brings too much new knowledge.
>
> **A1.** As discussed in Section 1 Introduction, warp-based image animation techniques such as First Order Motion Model are well-known to struggle at producing realistic images when there are large pose differences. Section 3.1 discusses exactly why vanilla CutMix is not directly applicable to image animation and we have demonstrated in our quantitative ablation study (Section 4.3: Table 4) that it sacrifices pose (AKD) and identity (AED) to pursue pixel realism. Our additional qualitative ablation study (Section 4.3: Figure 5) on vanilla CutMix suggests that vanilla CutMix struggles in distinguishing between foreground and background around locations with large motions. We have also revised Section 5 Discussion to reflect the key messages accordingly.
>
> **Q2. The choice of k**
>
> > In equation 2, there is the top-k number. Which k do you choose in your experiments?
>
> **A2.** We followed the official implementation of CutMix to randomize the value of k (lambda in their code) for each regularization and do not use a fixed k in our experiments. We have clarified this in Section 3.2 accordingly.
>
> https://github.com/clovaai/CutMix-PyTorch/blob/2d8eb68faff7fe4962776ad51d175c3b01a25734/train.py#L230
>
> **Q3. Evaluation on the choice of k**
>
> > I do see an ablation study in supp but it seems to be an evaluation of masked PSNR and SSIM, not the choice of k. I think it would be helpful to add this.
>
> **A3.** We disagree. Since the amount of occlusion varies for each training sample, we believe fixing k in the experiments would harm the generalizability instead of helping it.
>
> **Q4. Generalization**
>
> > To be specific, does top-k generalize CutMix?
>
> **A4.** Yes, top-k generalizes CutMix due to its flexibility in shape (irregular in top-k v.s. rectangular in CutMix) and location (arbitrary in top-k v.s. single region in CutMix).
>
> **Q5. The value of k and CutMix**
>
> > For example, if K = 0, it is CutMix?
>
> **A5.** No, top-k is CutMix only if there is a single region of occlusion and the region is in a rectangular shape.
>
> **Q6. Ablation study**
>
> > Inadequate ablation study
>
> **A6.** We additionally evaluated vanilla CutMix in our qualitative ablation study (Section 4.3) and have clarified the purposes of individual components accordingly.

---

### Official Review · AnonReviewer3 · 2020-10-29
**Both the contribution and the improvement are not significant**

**Rating:** 4
**Confidence:** 3

**Review:**

This paper proposes PriorityCut, a data augmentation technique for self-supervised image animation. It uses the top-k percent occluded pixels of the foreground for consistency regularization. By mixing real and fake images according to the proposed method, the generator and discriminator are encouraged to focus on difficult parts, thus improving results. Experiments show that the proposed method could provide mild improvements in some cases.

**Strengths**
+ The proposed PriorityCut is simple to implement and could help in some cases.

**Weaknesses**
- The idea is not significant. The overall framework is based on FOMM (Siarohin et al. 2019). It merely proposes a simple way to identify difficult pixels and have the generation module focus on the more difficult parts. The mixing operation is similar to CutMix (Yun et al. 2019). Different from CutMix that is demonstrated useful for many important computer vision tasks, the proposed data augmentation technique is designed and demonstrated only for a specific application, image animation.

- The improvements are very subtle. Although the tables show mild quantitative improvements, the gains are not significant. The visual enhancements compared to the baseline method are not substantial, either. For most examples in the supplementary document, the improvements against the baseline are very subtle. Also, the paper motivates the approach by occlusions, but the places where the proposed method helps to improve do not always exhibit occlusions.

- The paper is not well-written. For example, since both the binary background mask and the occlusion map are binary masks, I assume that the mask $\hat{\mathcal{O}}_{fg}$ is binary. It is not clear how to obtain the PriorityCut mask by taking the top k percent occluded pixels. It would be better to use a symbol for the PriorityCut mask, rather than the confusing operation $\min_k \hat{\mathcal{O}}_{fg}$.


**Minor issues**
- The paper mentions that the red and green colors indicate the worse and better result than the baseline. However, in Tables 1, 2, and 3, most of the numbers are not colored.

- The topic of the paper is about animating images. As the abstract states, the result is a video of a source image following a motion of a driving video. For video, in addition to image quality, it is essential to investigate its temporal coherence quality. Without seeing the videos, it is difficult to see how temporally coherent the resultant videos are.

**Post-rebuttal**

After reading the rebuttal and the other reviews, my rating remains the same, although the rebuttal addresses some issues. The added videos show good temporal coherence as previous methods, and several writing issues have been resolved. However, I still feel that the contribution of the proposed method is not significant enough. It would be more convincing if the paper can show that the proposed augmentation technique is effective for more applications. Also, I still think that the visual improvement is very subtle. For the six added videos, only fig_a2_label.mp4 shows clear improvement visually.

---

> ### Author Response · Authors · 2020-11-19
> **Comments for Review 3 (parts 1/2)**
>
> **Q1. Novelty**
>
> > The idea is not significant. The overall framework is based on FOMM (Siarohin et al. 2019). It merely proposes a simple way to identify difficult pixels and have the generation module focus on the more difficult parts. The mixing operation is similar to CutMix (Yun et al. 2019).
>
> **A1.** As discussed in Section 1 Introduction, warp-based image animation techniques such as First Order Motion Model are well-known to struggle at producing realistic images when there are large pose differences. Section 3.1 discusses exactly why vanilla CutMix is not directly applicable to image animation and we have already demonstrated in our quantitative ablation study (Section 4.3: Table 4) that it sacrifices pose (AKD) and identity (AED) to pursue pixel realism. Our additional qualitative ablation study (Section 4.3: Figure 5) on vanilla CutMix suggests that vanilla CutMix struggles in distinguishing between foreground and background around locations with large motions. We have also revised Section 5 Discussion to reflect the key messages accordingly.
>
> **Q2. Applications**
>
> > Different from CutMix that is demonstrated useful for many important computer vision tasks, the proposed data augmentation technique is designed and demonstrated only for a specific application, image animation.
>
> **A2.** Thank you for the suggestion. Due to time and resource constraints, it is not feasible to validate PriorityCut on other problems within the rebuttal period. However, we have considered experimenting PriorityCut on image inpainting for irregular holes, also known as free-form image inpainting. If time is allowed, we will be incorporating our findings into the results. Otherwise, we will be adding our analysis in future work.
>
> **Q3. Significance of Improvements**
>
> > The improvements are very subtle. Although the tables show mild quantitative improvements, the gains are not significant. The visual enhancements compared to the baseline method are not substantial, either. For most examples in the supplementary document, the improvements against the baseline are very subtle.
>
> **A3.** The degree of improvement is consistent with state-of-the-art techniques in both image animation and image inpainting literature published in top machine learning conferences.
>
> **Image Animation**
>
> For image animation on VoxCeleb, PriorityCut improved the L1 distance by 0.0012 compared to FOMM but Tripathy et al., 2021 [1] worsened the MSE (which also measures pixel distances) by 0.006 compared to FOMM and is the same as X2Face. PriorityCut improved the PSNR by 0.17 while Yao et al., 2020 [2] improved the PSNR by 0.212. PriorityCut improved both identity and pose but Burkov, Egor, et al., 2020 [3] showed a trade-off between identity and pose. For image animation on Tai-Chi-HD, PriorityCut improved the AKD by 0.09 compared to FOMM while Jeon, Subin, et al., 2020 [4] worsened the AKD by 1.27 compared to FOMM and is only on par with MonkeyNet.
>
> **Image Inpainting**
>
> For image inpainting, PriorityCut improved the PSNR and the SSIM in the ranges of 0.15 - 0.5 and 0.001 - 0.005, respectively. Liu, Guilin, et al., 2018 [5] improved the PSNR and the SSIM in the ranges of 0.1 - 0.55 and 0 - 0.008, respectively. Zheng, Chuanxia, et al., 2019 [6] improved the PSNR by 0.74. Xiong, Wei, et al., 2019 [7] improved the PSNR by 0.6 and the SSIM by 0.003. Xie, Chaohao, et al., 2019 [8] improved the PSNR and the SSIM in the ranges of 0.19 - 0.42 and 0.002 - 0.007, respectively.
>
> For visualization, we have included videos corresponding to the figures in the paper with noticeable improvements in the supplementary materials.
>
> [1] Tripathy, Soumya, Juho Kannala, and Esa Rahtu. "FACEGAN: Facial Attribute Controllable rEenactment GAN." Proceedings of the Workshop on Applications of Computer Vision. 2021.
>
> [2] Yao, Guangming, et al. "Mesh Guided One-shot Face Reenactment Using Graph Convolutional Networks." Proceedings of the 28th ACM International Conference on Multimedia. 2020.
>
> [3] Burkov, Egor, et al. "Neural Head Reenactment with Latent Pose Descriptors." Proceedings of the IEEE/CVF Conference on Computer Vision and Pattern Recognition. 2020.
>
> [4] Jeon, Subin, et al. "Cross-Identity Motion Transfer for Arbitrary Objects through Pose-Attentive Video Reassembling." Proceedings of the 16th European Conference on Computer Vision. 2020.
>
> [5] Liu, Guilin, et al. "Image inpainting for irregular holes using partial convolutions." Proceedings of the European Conference on Computer Vision (ECCV). 2018.
>
> [6] Zheng, Chuanxia, Tat-Jen Cham, and Jianfei Cai. "Pluralistic image completion." Proceedings of the IEEE Conference on Computer Vision and Pattern Recognition. 2019.
>
> [7] Xiong, Wei, et al. "Foreground-aware image inpainting." Proceedings of the IEEE conference on computer vision and pattern recognition. 2019.
>
> [8] Xie, Chaohao, et al. "Image inpainting with learnable bidirectional attention maps." Proceedings of the IEEE International Conference on Computer Vision. 2019.

---

> ### Author Response · Authors · 2020-11-19
> **Comments for Review 3 (parts 2/2)**
>
> **Q4. Locations of improvements**
>
> > The paper motivates the approach by occlusions, but the places where the proposed method helps to improve do not always exhibit occlusions.
>
> **A4.** We respectfully disagree with this. Focusing on occlusions, the difficult parts, does not necessarily imply that non-occluded parts cannot be improved or have to worsen. PriorityCut, being a supervised approach on occlusion, improved both salient and non-salient parts (Tables 1, 2, and 4), and both top-k percent and non-top-k percent occluded pixels (Tables A2, A3, A4). In contrast, unsupervised approaches (adversarial training and vanilla CutMix) sacrifice important image animation properties such as foreground realism, pose (AKD) and identity (AED) to pursue pixel realism. The experimental results demonstrate the effectiveness of PriorityCut in improving all-round realism instead of exhibiting trade-offs like previous approaches.
>
> **Q5. Presentation**
>
> > The paper is not well-written. For example, since both the binary background mask and the occlusion map are binary masks, I assume that the mask $\hat{\mathcal{O}}_{fg}$ is binary. It is not clear how to obtain the PriorityCut mask by taking the top k percent occluded pixels.
>
> **A5.** The occlusion map is an alpha mask, not a binary mask. This is mentioned in Section 3.2 and shown in Figures 2. Equation 1 uses the binary background mask to retain the foreground portion of the occlusion map shown in Figure 2, which is also an alpha mask. The $\min\limits_{k}$ operation sets the individual values of a mask to 0 (fully occluded) when it is within the k-th percentile and to 1 (not occluded) when it is outside the kth percentile.
>
> **Q6. Notation**
>
> > It would be better to use a symbol for the PriorityCut mask, rather than the confusing operation $\min\limits_{k}\hat{\mathcal{O}}_{fg}$.
>
> **A6.** As suggested, we replaced the $\min\limits_{k}$ operation for the PriorityCut mask with a symbol and have revised Section 3.2 accordingly.
>
> **Q7. Coloring scheme**
>
> > The paper mentions that the red and green colors indicate the worse and better result than the baseline. However, in Tables 1, 2, and 3, most of the numbers are not colored.
>
> **A7.** We applied red and green colors to only variants of the baseline (adversarial model, CutMix model, and ours) and only if they are not the best and the second-best results. We have clarified this in Section 4.2.
>
> **Q8. Temporal coherence**
>
> > The topic of the paper is about animating images. As the abstract states, the result is a video of a source image following a motion of a driving video. For video, in addition to image quality, it is essential to investigate its temporal coherence quality. Without seeing the videos, it is difficult to see how temporally coherent the resultant videos are.
>
> **A8.** As suggested, we have included videos corresponding to the figures in the paper in the supplementary materials for evaluation.

---

### Official Review · AnonReviewer4 · 2020-10-30
**An extension of first-order motion model for image animation with marginal improvement on visual quality**

**Rating:** 5
**Confidence:** 4

**Review:**

This paper proposes an extension of first-order motion model for image animation with a driving video by incorporating the U-net based discriminator from Schonfeld et al. The main contribution of this paper is their training strategy of generative network that mixes synthetic pixels with real pixels around the occlusion area where warping artifacts are more likely to appear. Results and experiments are reported on three diverse datasets: VoxCeleb, BAIR, and Tai-Chi-HD with comparisons to closely related works.

+This paper is well written and easy to read. Most of sections are clearly presented with sufficient details.
+Both qualitative and quantitative experiments are reported on three diverse datasets with solid baselines including the base methods: two variants of first-order motion models.

-The main concern is its novelty. The paper is mostly built upon the first-order motion model with the adoption of previously published U-net discriminator. The proposed pixel mixing method for generative network training is interesting, but it doesn't show significant improvement over the baseline. One way to further validate the proposed idea could be doing general image inpainting problems.
-It'd be better to include video results for evaluation.

---

> ### Author Response · Authors · 2020-11-19
> **Comments for Review 4**
>
> **Q1. Novelty**
>
> > The main concern is its novelty. The paper is mostly built upon the first-order motion model with the adoption of previously published U-net discriminator. The proposed pixel mixing method for generative network training is interesting, but it doesn't show significant improvement over the baseline.
>
> **A1.** The degree of improvement is consistent with state-of-the-art techniques in both image animation and image inpainting literature published in top machine learning conferences.
>
> **Image Animation**
>
> For image animation on VoxCeleb, PriorityCut improved the L1 distance by 0.0012 compared to FOMM but Tripathy et al., 2021 [1] worsened the MSE (which also measures pixel distances) by 0.006 compared to FOMM and is the same as X2Face. PriorityCut improved the PSNR by 0.17 while Yao et al., 2020 [2] improved the PSNR by 0.212. PriorityCut improved both identity and pose but Burkov, Egor, et al., 2020 [3] showed a trade-off between identity and pose. For image animation on Tai-Chi-HD, PriorityCut improved the AKD by 0.09 compared to FOMM while Jeon, Subin, et al., 2020 [4] worsened the AKD by 1.27 compared to FOMM and is only on par with MonkeyNet.
>
> **Image Inpainting**
>
> For image inpainting, PriorityCut improved the PSNR and the SSIM in the ranges of 0.15 - 0.5 and 0.001 - 0.005, respectively. Liu, Guilin, et al., 2018 [5] improved the PSNR and the SSIM in the ranges of 0.1 - 0.55 and 0 - 0.008, respectively. Zheng, Chuanxia, et al., 2019 [6] improved the PSNR by 0.74. Xiong, Wei, et al., 2019 [7] improved the PSNR by 0.6 and the SSIM by 0.003. Xie, Chaohao, et al., 2019 [8] improved the PSNR and the SSIM in the ranges of 0.19 - 0.42 and 0.002 - 0.007, respectively.
>
> **References**
>
> [1] Tripathy, Soumya, Juho Kannala, and Esa Rahtu. "FACEGAN: Facial Attribute Controllable rEenactment GAN." Proceedings of the Workshop on Applications of Computer Vision. 2021.
>
> [2] Yao, Guangming, et al. "Mesh Guided One-shot Face Reenactment Using Graph Convolutional Networks." Proceedings of the 28th ACM International Conference on Multimedia. 2020.
>
> [3] Burkov, Egor, et al. "Neural Head Reenactment with Latent Pose Descriptors." Proceedings of the IEEE/CVF Conference on Computer Vision and Pattern Recognition. 2020.
>
> [4] Jeon, Subin, et al. "Cross-Identity Motion Transfer for Arbitrary Objects through Pose-Attentive Video Reassembling." Proceedings of the 16th European Conference on Computer Vision. 2020.
>
> [5] Liu, Guilin, et al. "Image inpainting for irregular holes using partial convolutions." Proceedings of the European Conference on Computer Vision (ECCV). 2018.
>
> [6] Zheng, Chuanxia, Tat-Jen Cham, and Jianfei Cai. "Pluralistic image completion." Proceedings of the IEEE Conference on Computer Vision and Pattern Recognition. 2019.
>
> [7] Xiong, Wei, et al. "Foreground-aware image inpainting." Proceedings of the IEEE conference on computer vision and pattern recognition. 2019.
>
> [8] Xie, Chaohao, et al. "Image inpainting with learnable bidirectional attention maps." Proceedings of the IEEE International Conference on Computer Vision. 2019.
>
> **Q2. Applications**
>
> > One way to further validate the proposed idea could be doing general image inpainting problems.
>
> **A2.** Thank you for the suggestion. Due to time and resource constraints, it is not feasible to validate PriorityCut on other problems within the rebuttal period. However, we have considered experimenting PriorityCut on image inpainting for irregular holes, also known as free-form image inpainting. If time is allowed, we will be incorporating our findings into the results. Otherwise, we will be adding our analysis in future work.
>
> **Q3. Video Results**
>
> > It'd be better to include video results for evaluation.
>
> **A3.** As suggested, we have included video results corresponding to the figures in the paper in the supplementary materials for evaluation.

---

### Author Response · Authors · 2020-11-19
**Review Summary**

We thank the reviewers for their insightful and thoughtful feedback. We are encouraged that they find PriorityCut an interesting idea outperforming state-of-the-art (**R2**), evaluated both qualitatively and quantitatively on three diverse datasets with solid baselines (**R4**), simple to implement, and could help in some cases (**R3**). Reviewers found our paper well written and easy to read, and most of the sections are clearly presented with sufficient details (**R4**). The primary concerns are novelty and the significance of improvements. We address reviewer comments below and will incorporate all feedback.

---

### Decision · Program_Chairs · 2021-01-07
**Final Decision**

**Decision:**

Reject

**Comment:**

All the reviewers agreed that the paper lacks novelty. The overall framework is based on FOMM (Siarohin et al. 2019); the mixing operation is similar to CutMix (Yun et al. 2019). The improvements over the prior work are very subtle. R1 and R3 mentioned the paper is not well-written. The rebuttal didn’t change the reviewers’ mind. In particular, the reviewers pointed out that only one out of six videos showed clear visual improvement in the added video results. After reading the paper, reviewer’s comments, the rebuttal with added results, the AC agrees with the reviewers that the paper is not ready for publication.